# First and Later Dating Experiences and Dating Violence among High School Students

**DOI:** 10.3390/ijerph20064870

**Published:** 2023-03-10

**Authors:** Sigita Lesinskienė, Natalja Istomina, Greta Stonkutė, Jelizaveta Krotova, Rokas Šambaras, Donatas Austys

**Affiliations:** 1Clinic of Psychiatry, Institute of Clinical Medicine, Faculty of Medicine, Vilnius University, 01513 Vilnius, Lithuania; 2Institute of Health Sciences, Faculty of Medicine, Vilnius University, 01513 Vilnius, Lithuania; 3Faculty of Medicine, Vilnius University, 01513 Vilnius, Lithuania

**Keywords:** dating, high school students, adolescents, dating violence, sexual harassment, alcohol use, drug use, inebriation, sex education, cultural contexts

## Abstract

This study investigated early dating experiences by exploring the complex aspects of first-time and later romantic involvement as well as the surrounding circumstances. In total, 377 young people (with the median age being 17 years) were investigated in six high schools in two cities in Lithuania using a questionnaire designed for research purposes by the authors. The results from this study in Lithuania contribute to the field by showing current data on the cultural and psychosocial aspects of dating experiences in high schools. This investigation into first-time and later dating creates an appropriate window to explore and investigate the attitudes, dating habits, and experiences of late adolescents, comprising negative experiences and sexual harassment, which could be used to build preventative programmes. The obtained results provide various data that are useful for trying to better understand the current habits and experiences of young people, for public health specialists, educators, and doctors and also provide an opportunity for monitoring trends, dynamic changes over time, and cross-cultural comparisons.

## 1. Introduction

First close friendships, first meetings, falling in love, and dating are important memories and experiences in a young person’s life. There are relatively few scientific data on this topic, though there has been growing interest in recent years. While dating, young people experience personal, exciting, and positive experiences, but some face unpleasant or even dangerous and harmful experiences. Dating violence, sexual harassment, alcohol abuse, drug abuse, and other types of risk-taking behaviour in adolescents could lead to social maladjustment and distorted attitudes towards building personal relationships and towards violence. Previous research has shown that dating violence negatively impacts adolescents’ mental health outcomes and suicidal behaviours [1,2,3,4]. The systematic review revealed that adolescent dating violence is highly prevalent and can have serious health consequences, including homicides, and be a predictor of intimate partner violence in adulthood [5]. A literature review of school bullying and dating violence in 12–19 years old adolescents pointed out that involvement in bullying perpetration or Victimization could be risk factors for perpetration or Victimization in early romantic relationships that emerge within an evolving peer group [6].

Prevailing styles, trends, and dating tendencies can vary across countries and change over time, which is important for the field of public health. Investigations into dating experiences, expectations, and attitudes among young people are necessary. Research data give a broader view of developmental trends, provide background to preventative programmes, and format adequate policy and mental health practice. Research findings suggested that schools, practitioners, and policy-makers do not meet the needs of adolescents in regards to dating violence prevention and intervention and highlighted the importance of talking about dating violence as a public health issue among adolescents [7,8].

However, little research is available on first dating experiences. Previous work mostly focused on first sexual intercourse experiences; however, most of it failed to address other aspects of early romantic relationships. The first dating topic overlaps with the public health, medical, social, and psychological fields. Since building a romantic relationship is new for adolescents, their dates and romantic relationships differ from those of adults.

In this study, we were specifically interested in including complex aspects of first-time dating and later romantic involvement as well as the surrounding circumstances. It is essential to highlight that early dating is not only different from later dating but also very sensitive and unique. Romantic relationships constitute a new dimension in an adolescent’s social life and are important to their well-being. A romantic debut can be a traumatic or nurturing experience, leaving a lasting impression. Scientific publications showed the influence of dating experience on adolescents’ emotions and expectations and, in the long run, on romantic and sexual experiences in adulthood [9,10]. A healthy start and healthy relationships could be a resource of well-being; teenagers themselves note that their romantic partners are a strong source of support and intimacy [11]. Research showed that romantic relationships are a predictor of psychological well-being, having a positive link with positive interpersonal relationships and life development and a negative link with autonomy and self-acceptance [12]. A recent study’s findings suggested a great variability in adolescents’ romantic relationship experiences and pointed towards the developmental significance of these experiences for short- and long-term well-being [13].

Our study aimed to investigate the opinions and attitudes of young people in high school on dating and their own experiences.

## 2. Materials and Methods

### 2.1. Participants and Procedure

Acknowledging the lack of scientific data on teenagers dating, this survey was organized on an initiative and voluntary basis in close cooperation with the Vilnius University Faculty of Medicine and the Ministry of Education. The two largest cities in Lithuania were selected for the study. One of them was Vilnius, which is the capital of Lithuania. According to the Lithuanian Statistics Department, Vilnius has a population of 581,475, and there are 73,883 students from 1st to 12th grade. Klaipėda is the largest city in western Lithuania. According to the Statistics Department of Lithuania, Klaipėda has a population of 152,237, and there are 21,267 students from 1st to 12th grade. Six schools in two Lithuanian cities, Vilnius and Klaipeda, were invited, and both agreed to participate in the anonymous survey. Permissions from the communities and school administrations were obtained. High school students in grades 11 and 12 were interviewed through an anonymous questionnaire. Participants were informed about the aims and scope of the survey. The only criterion for inclusion was the age of the students, who had to be over 16 years old. All participants filled out the paper questionnaires of their own free will. Researchers went to schools and talked with the adolescents before they filled in the questionnaires. The study aims and purposes were described to the teenagers, and they were invited to anonymously and voluntarily join the research project. Following this introductory part, all students agreed to complete the questionnaire. Some of the respondents were missing answers to some questions, but, generally, we observed active involvement and interest in the research subject among the participants.

### 2.2. Questionnaire

The original anonymous questionnaire was developed for research purposes by authors and consisted of 45 items divided into nine sections, which were designed and titled in order to provide more clarity and visual structure for the respondents. The questions were divided into separate sections and comprised socio-demographic data (age; Gender: female/male; place of residence: Vilniu/Klaipėda; living with parents: both parents/just my mother/just my father/with grandparents/with mom and stepfather/with dad and stepmother/with parents who adopted me/with guardians), sexual orientation: heterosexual/bisexual/homosexual/asexual, questions about first and later dating experiences and circumstances, number and durations of close friendships, attitudes, safety aspects, who they meet to talk about dating and friendships, what age is most relevant to start dating, and opinions and needs regarding sex education in school.

### 2.3. Statistical Analysis

The statistical analysis (comparisons) was performed according to the background variables. The main background variables included gender, sexual orientation, place of residence, family status, and ethnicity of parents. Additional variables included dating unknown persons and dating with persons from social networks. In order to simplify the interpretation of the results of the statistical tests and avoid groups with low numbers of respondents, all background variables were transformed to binary.

The background variables were used for crosstabulations with such variables as dating experience (having been to at least one date vs. having not been to any date) and the presence of any type of harassment (negative vs. non-negative dating experience). Responses where verbal, physical, or sexual harassment was indicated were defined as negative dating experiences, and other responses (“I do not know” or “No”) were defined as non-negative dating experiences. The normality of the variables’ distribution was tested using the Shapiro–Wilk test, and there were no variables with normal distribution. Thus, medians with interquartile range (Q1–Q3) were presented for variables with non-normal distribution. Pearson’s chi-squared test (χ^2^) was used to determine whether there was a statistically significant difference between the expected frequencies and the observed frequencies in one or more categories. If more than 20% of expected values were lower than 5, Fisher’s exact test was used instead. Additionally, binary logistic regression analysis was performed in order to control for the background variables during the assessment of the association between background variables with the presence of dating experience and the presence of harassment during the dates. Adjusted odds ratios were calculated for each statistically significant variable of the model. *p*-value of the Hosmer–Lemeshow goodness of fit test and Negelkerke R Square values were presented in order to show the characteristics of the model. Differences were considered statistically significant when *p* < 0.05.

IBM SPSS Statistics (version 22) and Microsoft Excel 2016 were used to analyze the data.

## 3. Results

### 3.1. Main Characteristics of the Sample

This study included 377 adolescents with a median age of 17 (17–18) years. Males and females were similarly represented in the sample. The majority of the respondents indicated living with both parents and having a heterosexual orientation. Respondents from Vilnius comprised a larger part in the sample. None of the sociodemographic factors were related to the presence of dating experience among the participants of this study (*p* > 0.05) (Table 1).

### 3.2. Non-Negative and Negative Dating Experiences among Adolescents Who Were Dating

The majority, 184 (61.7%), of the respondents who were dating (have more dating experience than one date) indicated that they dated persons they did not know. A large part, 153 (51.3%), of the respondents who were dating met with persons from social networks.

The median age of the first date of the respondents was 14 (13–16) years. Girls started dating older than boys: 15 (14–16) years and 14 (13–15) years, respectively (*p* = 0.009). Moreover, individuals with a non-heterosexual orientation started dating when older: 15 (14–16) years vs. 14 (13–15) years, respectively (*p* = 0.013). In addition, the median age of the first date was greater among those who met with a person not from social networks: 15 (14–16) years vs. 14 (13–15) years, respectively (*p* = 0.031). The age of the first date was not associated with place of residence, family status, or meeting with unknown persons (*p* > 0.05).

The majority 215 (72.3%) of the respondents who were dating indicated the first date was with a known person, whereas 13.9% indicated that they met with a person they did not know. The distribution of the respondents for the persons they met with was similar according to gender, sexual orientation, place of residence, and family status (*p* > 0.05). The majority of the respondents who were dating indicated the public place of the first date: 197 (66.1%) indicated a walk, 68 (22.8%)—a cafe, 45 (15.1%)—a cinema, 6 (2.0%)—a bar or club, 3 (1.0%)—a theatre. Only a few of the respondents (2.0%) indicated that their first date was at home. Dating at home was significantly more frequent in Klaipeda than in Vilnius: 5.3% vs. 0.5%, respectively (*p* = 0.013). Gender, sexual orientation, place of residence, family status, meeting with unknown persons, or meeting with persons from social networks was not associated with dating at home for the first time (*p* > 0.05).

Most, 182 (61.5%), of the respondents who were dating indicated that the first date was initiated by the persons they met with, whereas 99 (33.4%) indicated that they initiated the dates themselves. Boys, significantly more often than girls, indicated they initiated the first date (70.6% vs. 2.1%, respectively, *p* < 0.001). The distribution of the respondents for this question was not associated with sexual orientation, place of residence, or family status (*p* > 0.05).

The majority, 163 (55.1%), of the respondents who were dating indicated that the duration of the first date was from 1 to 3 h, 105 (35.5%) of the respondents indicated that their first date lasted the whole evening, and 15 (5.1%) of the respondents indicated that their first date lasted the whole evening and night. The distribution of the respondents for the duration of their first date was similar with respect to gender, sexual orientation, place of residence, and family status (*p* > 0.05).

A significant part, 52 (17.4%), of the sample indicated that they faced harassment during the date: 23 (7.7%) of the respondents noted verbal harassment, and 32 (10.7%) of the respondents indicated physical or sexual harassment. A few, 3 (1.0%), respondents indicated experiencing verbal and physical harassment. Other than heterosexual orientation, dating unknown persons was associated with relatively more frequent verbal or physical harassment (*p* < 0.05). Gender, place of residence, family status, and parental ethnicity were not associated with the experience of harassment (*p* > 0.05). The distributions of the respondents according to their positive and negative dating experiences are presented in Table 2.

Among those who experienced verbal or physical harassment, one-third, 16 (30.8%), of the respondents indicated that they or their partners were under the effect of psychotropic substances. This was not associated with gender, family status, dating unknown persons, or dating persons from social networks (*p* > 0.05). On the other hand, it was associated with place of residence: being under the effect of psychotropic substances was indicated by 14 (42.4%) of the respondents from Vilnius and 2 (9.1)% of the respondents from Klaipėda (*p* = 0.016).

### 3.3. Talking about Dating Experiences

Half, 142 (51.3%), of the respondents indicated that they often talk about their dates with friends. Talking about friends’ dates was indicated by 159 (44.5%) of the respondents. A significantly smaller part, 70 (24.9%), of the respondents indicated often talking about their dates with family members. Only one (0.4%) respondent indicated often talking about their dates with teachers. Few, 4 (1.4%), respondents indicated talking about their dates in social networks (Table 3). The distribution of the respondents regarding the frequency of talking about their dates was not associated with place of residence or family status (*p* > 0.05). Respondents with a non-heterosexual orientation more frequently indicated often talking about their dates (*p* = 0.039). Sexual orientation was not associated with talking about dates with family members, with teachers, or in social networks, and there were similar results for talking with friends about the friends’ dates (*p* > 0.05). Girls, more often than boys, indicated talking about their dates with friends and family members, including talking with friends about the friends’ dates (*p* < 0.05). Talking about dates with teachers or in social networks was not associated with gender (*p* > 0.05).

The majority, 217 (57.6%), of the respondents indicated that in the case of a negative experience during a date, they would talk about that with their friends. Talking about negative experiences with their mother, father, brother or sister, or another important adult was indicated by 165 (43.8%), 80 (21.2%), 80 (21.2%), and 32 (8.5%) of the respondents, respectively. Talking about negative experiences with other persons, including physicians, grandparents, teachers, social workers, and psychologists, as well as talking about negative experiences in social networks or helplines was indicated by less than 5% of the sample. A significant part, 38 (10.1%), of the sample noted that they would not talk with anybody about their negative dating experiences. Girls indicated that they would talk about their negative experiences more frequently with their mothers, their brothers or sisters, and other important adults, whereas boys and adolescents with other than heterosexual orientation indicated that they would talk about their negative experiences more frequently with friends (*p* < 0.05). Place of residence was not associated with the choice of persons to speak with about negative dating experiences.

### 3.4. Having a Partner

A quarter 118 (31.3%) of the respondents indicated having a partner; a large proportion 44 (37.3%) of them indicated having a partner from their school environment, whereas the majority 74 (62.7%) of them indicated having a partner from a different environment than school.

The vast majority, 261 (94.9%), of the respondents noted having had up to six partners in their life, whereas 190 (69.1%) of the respondents indicated having had one partner in their life. The number of partners was greater among respondents from Klaipeda (*p* < 0.001). Gender, sexual orientation, and family status were not associated with the number of partners (*p* > 0.05).

Half, 111 (48.1%), of the respondents indicated that the most appropriate age to start dating and relationships is from 15 to 17 years, whereas 9 (3.9%) respondents indicated the most appropriate age to start dating and relationships is from 10 to 14 years. Girls, more frequently than boys, indicated an older age (*p* = 0.003). Sexual orientation and family status were not associated with the difference in opinions about the most appropriate age to start dating and relationships.

### 3.5. Dating Expectations among Adolescents without Dating Experience

Among the adolescents with no dating experience, the majority, 31 (42.5%), indicated that they do not want to date yet, 12 (16.4%) indicated that they would look for a partner in social networks, 3 (4.1%) indicated that they would look for a partner at school, and 27 (37.0%) indicated that they would look for a partner in other places than at school or in social networks.

The majority, 35 (50.7%), of the adolescents with no dating experience indicated that there is no such person they would like to date, 23 (33.3%) indicated that there is such a person, and the rest, 11 (15.9%), of the respondents did not know whether there is such a person.

Among those who indicated that there is a person they would like to date, one-third, 9 (33.3%), indicated that they are afraid to invite such a person, a similar part, 8 (29.6%), indicated that they are waiting to be invited by such a person, 6 (22.2%) indicated that they do not know how to invite such a person, and 4 (14.8%) indicated other reasons.

When asked about the date setting, only one (1.4%) adolescent with no dating experience indicated the wish to date at home. Other respondents indicated public places: a walk, a cafe, a cinema, a bar, a theatre, and other places (respectively, 54.2%, 13.9%, 8.3%, 8.3%, 4.2%, and 9.7%).

## 4. Discussion

Our study revealed that the median age for the first date was 14 years. This finding is a new attempt to obtain such data and will help build preventative programmes in schools at the appropriate age. Most of the respondents (with the median age being 17 years) already had dating experience. The respondents of our study were high school students in grades 11 and 12 who fit into the group of late adolescents. The three places for the first date (walk in the park or city, cafe, and cinema) were most often mentioned by late adolescents and could be considered as safe and popular environments. Most of the respondents indicated that they went on a first date with someone they knew and liked. The duration of the first date was typically several hours; only 5.1% noted that their first date lasted the whole evening and night. In addition, 24.4% of the respondents indicated having a partner currently, and the majority of the respondents, 62.7%, noted that their partner is from an environment other than school.

The proportion of those dating at home differed in the two cities. Our study revealed that some tendencies related with dating, substances use could vary even across a small country, which could be an important message for schools and public health practitioners.

Interestingly, the data revealed the still-prevailing tendency that boys, significantly more often than girls, indicated initiating the first date. Boys made the first date invitation more often than girls and started dating earlier. Among individuals with a non-heterosexual orientation, the mean age of the first date was 15 years. Brouwer et al. [14], in their recent study using European data, also revealed that individuals with differences in sex development were less likely to reach each of the romantic and sexual milestones compared to their peers without these conditions, and they were significantly older when reaching such milestones. The results of our study revealed that for those respondents who indicated a non-heterosexual orientation, dating unknown persons was associated with relatively more frequent verbal or physical harassment.

There are surprisingly few or even no data available on first dating among adolescents that explore common dating tendencies that are not related to drug abuse or violent behaviour. An investigation into dating trends among adolescents in schools could bring valuable information, which would help develop a better understanding of adolescents’ experiences and could serve as a relevant basis for preventative public health and early intervention programmes.

According to the literature review and our study results, sexual harassment is a common and widespread phenomenon [2,5]. Our results revealed that 17.4% of the respondents noted that they faced sexual harassment during the date: 7.7% was verbal, 10.7% was physical, and 1.0% was both types, and there were no associations found with gender, place of residence, family status, dating unknown persons, or dating persons from social networks. Recent studies highlighted the importance of investigating the attitudes towards dating violence among adolescents and emerging adults [15,16]. Courtain and Glowacz [15], in a Belgian study, found that males and younger participants show a higher tolerance towards every form of dating violence; participants are more tolerant towards male-perpetrated psychological dating violence than female-perpetrated psychological dating violence and are more tolerant towards female-perpetrated physical and sexual dating violence compared with male-perpetrated physical and sexual dating violence. It is important to investigate both dating experiences and the process of the formation of attitudes. Valdivia-Peralta et al. [16], in the province of Concepción, Chile, reported greater justifying attitudes towards violence in early adolescents than in late adolescents; in the authors’ comparison regarding sex, male adolescents tended to justify violence more than female adolescents. Furthermore, orientation towards future interventions was proposed in this study [16], and it is suggested that aspects related to sampling and possible modulating variables, such as cognitive development and moral development, should be considered by future investigations.

Cava et al. [17], in their study, investigated relations among romantic myths, offline dating violence victimisation, and cyber dating violence victimisation in adolescents. They revealed that verbal–emotional offline dating violence victimisation was the main predictor of cyber-control victimisation, and physical and relational offline dating violence victimisations were the main predictors of cyber-aggression victimisation. The authors suggested that these results can be useful for developing more effective offline and cyber dating violence-prevention programmes [17]. Further studies on these aspects could be planned and implemented, comprising cross-cultural comparisons and changes in the tendencies of the dynamic. According to this study’s data, no associations were found between sexual harassment and arrangements of dates in social networks.

The median age of the first date in our study was 14 years, which was greater among females and heterosexually oriented respondents than among males. The majority of the respondents had experience with romantic dating and partnership: 65.7% of the respondents indicated having one intimate friendship partner in their life. The number of partners were significantly greater in Klaipeda, indicating the importance of assuming differences between cities in the same country. Interestingly, gender, sexual orientation, and family status were not associated with the number of partners.

In this study, we obtained valuable answers from late adolescents that allowed us to better understand those young people who have not yet started dating. We did not find such research data or studies for any comparison. Therefore, future studies may try to expand the target population and further investigate this group of young people. Our research revealed that among respondents who had no dating experience, 33.3% indicated that there was a person they would like to date, 33.3% noted that they were afraid to invite that person for a date, 29.6% noted that they were waiting to be invited by that person, and 22.2% noted that they did not know how to invite that person on the date. Interestingly, when those with no dating experience were asked about the place to date, their answers demonstrated similar tendencies as the respondents with dating experience (walk, cafe, cinema, bar, and theatre).

All the respondents were asked about their opinion regarding the most appropriate age to start dating and romantic relationships, and, most often, the answers indicated an age from 15 to 17 years, though girls, significantly more frequently indicated an older age than boys. The results revealed that most often, the respondents were talking about their dating experiences with friends and then family members, but not with teachers, psychologists, social workers, or other school workers. This was an important finding that suggests that focusing prevention activities on peers and assuming that interventions shape the general understanding and emotional intelligence of adolescents at school could serve as a valuable resource of peer support, building good mental health for young people. An exploration of the dating expectations among late adolescents without dating experience revealed that 42.5% of them do not want to date yet, 50.7% noted that there is no such person they would like to date, and 33.3% indicated that there is such a person, but they are afraid to invite them on a date or are waiting to be invited by the person that they like; these aspects could also be included when building programmes and talking about dating in schools.

Spencer et al. [2], in a recently published meta-analysis on the risk markers for physical teen dating violence perpetration, highlighted the importance of the microsystem, specifically the dating relationship itself, when identifying adolescents at risk; the data revealed that on the ontogenetic level, externalising behaviours, approval of violence, risky sexual behaviours, alcohol use, depression, and delinquency were the strongest risk markers for teen dating violence perpetration. The data from our study demonstrated that among the respondents who indicated an experience of verbal or physical sexual harassment, 30.8% of them also noted that they or their partners were under the effect of psychotropic substances, which was not associated with gender, sexual orientation, family status, or dating unknown persons. Interestingly, it was related to the place of residence: 42.4% of the respondents from Vilnius and 10.5% of the respondents from Klaipeda. The findings of this study could be emphasised when elaborating on the more specific preventative public health programmes in different cities of the country.

Studies on the risk factors include examinations of childhood adversities, exposure to sexually explicit material and substance use, and the role of gender-inequitable attitudes in violence perpetration [18,19]. Important issues regarding the contextual risk profiles and trajectories of adolescent dating violence perpetration were discussed in a Reyes et al. study [20], with the results suggesting that early interventions to reduce violence exposure and increase social regulation across multiple social contexts may be effective in reducing dating violence perpetration across adolescence. The highest levels of psychological and physical perpetration across grades 8 through 12 were found among adolescents with a profile characterised by high levels of school, neighbourhood, and family risk [20].

In addition, Collibee et al. [21] found that adolescents involved in the juvenile justice system face a variety of risk factors associated with more frequent and severe experiences of aggression within romantic relationships compared to community samples. The authors also highlighted that risk for adolescent dating violence does not differ by offense type, suggesting that prevention efforts targeting adolescent dating violence at the earliest possible intervention point, regardless of first-time offense type or severity, may be especially impactful [21].

Lundgren and Amin [22] reviewed approaches to prevent adolescent intimate partner violence and sexual violence and identified critical knowledge gaps. They found that school-based dating violence interventions and community-based interventions to form gender-equitable attitudes among boys and girls show considerable success. The authors also found evidence that suggests that parenting interventions and interventions with children and adolescents subjected to maltreatment hold promise in preventing intimate partner violence or sexual violence by addressing child maltreatment, which is a risk factor. The results of that review suggested that programmes with longer-term investments and repeated exposure to ideas delivered in different settings over time have better results than single awareness-raising or discussion sessions. However, that review also pointed out that a lack of rigorous evidence limits the conclusions regarding the effectiveness of adolescent prevention programmes and indicated a need for more robust evaluation [22].

A global systematic review of evaluation studies synthesised the evidence from rigorous studies of prevention programmes for adolescent dating violence and found that, overall, 26 (50%) of the 52 evaluations reported a significant preventive effect on at least one outcome for adolescent dating violence; these findings suggested that research is needed to shed light on how adolescent dating violence-prevention programmes work and to identify whether such programmes’ effects are generalised across different settings, outcomes, and subgroups [23].

## 5. Study Limitations

The high school respondents in our study completed a questionnaire that focused on their experiences and views. The specific instruments and scales (the Myths of Romantic Love Scale, Conflict in Adolescent Dating Relationships Inventory, and Cyber-Violence in Adolescent Couples Scale) used in another recent study [17] were not implemented. Further studies in this field would broaden the scope of investigations. Research data demonstrated that adolescent dating violence is associated with suicide attempts [1,3,14]. Questions about suicidal thoughts and attempts were not included in the questionnaire of the present study. This could be investigated in further research. Given the high suicidal rates in Lithuania, the shortcomings of complex assistance [24,25,26], and the high rates of sexual harassment experiences in a pilot study in high school [27], it could be beneficial to plan a broader analysis of the psychosocial interconnections that affect adolescents’ mental health.

In the present study, we did not ask respondents about bullying, as we tried to keep the focus on the various aspects of dating and not burden high school students with a long questionnaire or difficult questions. A recent meta-analysis suggested that bullying and dating violence could be different behavioural manifestations, in different evolutionary moments and in different contexts, of the same underlying antisocial or violent dispositions, although longitudinal studies are needed to confirm this [6]. How dating violence relates to bullying and sexual harassment in high school remains a theme that needs broader exploration. The currently available research data showed that the links between bullying and dating violence in a high school population require further research [28]. Questions about self-esteem and body image could also be added to future research on the dating experiences of adolescents. In a recently published study, a moderated mediation analysis verified that low self-esteem is relevant in dating violence victimisation through body dissatisfaction, which is higher in women than in men [29].

### Ideas for the Implementation in Preventative Programmes

Public health specialists, clinicians, and educators can routinely include themes about dating experiences, bullying, substance abuse, and suicidal thoughts and organise preventative programmes in educational settings. Implementation discussions with adolescents about dating and safety should be carried out early enough, given that the median age of the first date of the study respondents was 14 years. A prevention programme should comprise topics about the first date; related views, imaginations, and experiences; and how adolescents could obtain help if they want to prepare and discuss the issues that concern them. The research data could be used and presented for those adolescents that want to discuss various related aspects of the possible experiences to prevent the risk of sexual harassment.

## 6. Conclusions

This investigation into late adolescents in high schools, focusing on questions about first and later dating, created an appropriate window to explore their attitudes, dating habits, and experiences. Researching the views of young people and the circumstances of their dating helps build prevention programmes for sexual harassment and negative experiences. The obtained results are essential for public health specialists, educators, and doctors when trying to understand the current habits and experiences of young people, which can shape preventative programmes in the most realistic and appropriate way, together with the provision of an opportunity for monitoring trends, dynamic changes over time, and cross-cultural comparisons.

## Figures and Tables

**Table 1 ijerph-20-04870-t001:** Distribution of the respondents regarding the presence of dating experience and sociodemographic factors.

Factor *	Total		With Dating Experience		Without Dating Experience		*p*-Value
	*n*	Relative Frequency (%) **	*n*	Relative Frequency (%)	*n*	Relative Frequency (%)	
**Gender**							
Male	185	49.1	142	76.8	43	23.2	0.284
Female	192	50.9	156	81.3	36	18.8	
**Sexual orientation**							
Heterosexual	323	87.1	257	79.6	66	20.4	0.692
Other	48	12.9	37	77.1	11	22.9	
**Place of residence**							
Vilnius	267	70.8	204	76.4	63	23.6	0.050
Klaipėda	110	29.2	94	85.5	16	14.5	
**Family status**							
Living with both parents	236	62.9	186	78.8	50	21.2	0.810
Other	139	37.1	111	79.9	28	20.1	
**Ethnicity of parents**							
Lithuanian	243	64.8	186	76.5	57	23.5	0.124
Other or multiethnic	132	35.2	110	83.3	22	16.7	

* Logistic regression analysis also revealed that none of these factors were associated with dating experience with and without control for all the rest of the factors mentioned in this table. ** Total relative frequency was counted within each column.

**Table 2 ijerph-20-04870-t002:** Distributions of the respondents regarding dating experience (presence of harassment) and socio-demographic factors.

Factor *	Non-Negative Dating Experience		Negative Dating Experience		*p*-Value
	*n*	Relative Frequency (%)	*n*	Relative Frequency (%)	
**Gender**					
Male	119	86.2	19	13.8	0.088
Female	121	78.6	33	21.4	
**Sexual orientation**					
Heterosexual	213	84.5	39	15.5	0.009
Other	24	66.7	12	33.3	
**Place of residence**					
Vilnius	165	83.3	33	16.7	0.459
Klaipėda	75	79.8	19	20.2	
**Family status**					
Living with both parents	149	82.3	32	17.7	0.914
Other	90	81.8	20	18.2	
**Ethnicity of parents**					
Lithuanian	152	82.2	33	17.8	0.956
Other or multiethnic	86	81.9	19	18.1	
**Met with persons they did not know**					
Yes	140	76.9	42	23.1	
No	94	90.4	10	9.6	0.005
**Met in social networks**					
Yes	119	78.3	33	21.7	
No	121	86.4	19	13.6	0.069

* Logistic regression analysis also revealed that controlling for all factors mentioned in this table, non-heterosexual orientation and meeting with unknown persons was associated with negative dating experiences (harassment) (odds ratios respectively were 2.3 and 4.3, *p* < 0.05). All other factors mentioned in this table were not associated with negative dating experience (*p* > 0.05). Negelkerke R Square was 0.098, *p*-value of Hosmer and Lemeshow Test was 0.661.

**Table 3 ijerph-20-04870-t003:** Distributions of the respondents regarding frequency of talking about dating.

Factor	Often		Rare		Never	
	*n*	Relative Frequency (%)	*n*	Relative Frequency (%)	*n*	Relative Frequency (%)
Talking about their dates with friends	142	51.3	110	39.7	25	9.0
Talking about their friends’ dates	159	44.5	133	37.3	65	18.2
Talking about their dates with family members	70	24.9	110	39.1	101	35.9
Talking about their dates with teachers	1	0.4	26	9.4	249	90.2
Talking about their dates in social networks	4	1.4	40	14.2	237	84.3

## Data Availability

The raw data supporting the conclusions of this article will be made available by the authors, without undue reservation.

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
