# Peer review of "First and Later Dating Experiences and Dating Violence among High School Students"

_ijerph, 2023, doi:10.3390/ijerph20064870_

Round 1

Reviewer 1 Report

The title of the reviewed article is: First and Later Dating Experiences and Dating Violence among School Graduates.

The study aimed to investigate the opinions and attitudes of young people in high school on dating and their own experiences. The authors conducted a cross-sectional survey in six Lithuanian schools. The respondents were 17-18 year old and they were asked about the experiences of first datings. The study presents important findings on this very little researched topic. It showed that adolescents in Lithuania start dating very early, at the age of 14 years. Furthermore, it indicated that these young people often had experienced negative events in terms of dating, including both physical and verbal violence. Most often they talk about the events among peers, and more seldomly to adults. These findings urge for development of preventive interventions that increase awareness of mental and sexual health, how to set own boundaries, what is appropriate treatment of one another and when to get help.

The authors have formulated a clear but rather broad aim. Specific objectives could be added as well as hypotheses they set out to test.

Generally manuscript is clearly written, very topical, interesting and important contribution to the field. The write-up could benefit of some restructuring and editing. Sub-headings in materials and methods could be bolded. It would also be reader-friendly if the sub-headings from the results section could somehow be integrated in the methods section, under the ‘questionnaires’. Also, the same sub-headings could be used in the discussion.

References seem relevant and mostly recent, with a few exceptation, mainly referring to particular scales. Number of authors´ own citations is moderate and the cited articles relevant.

The hypothesis was not stated and needs to be added. Cross-sectional study design should be indicated as well as the time period that the study was conducted. Were the data gathered during the pandemic? The students responded to the questionnaire only once?

Mostly, the results are presented clearly enough, however, I had some remarks and concerns about the presentation and with regard to some specific results (sexual/gender minority, place of residence), also interpretation.

Conclusions are mostly consistent with the presented evidence and arguments.

The study has an ethics approval from the Institutional Review Board at Vilnius University Faculty of Medicine and the authors state that they will make the  raw data available without undue reservation.

The question is novel and important with respect to adolescent mental health and provides useful guidelines and recommendations for preventive interventions. The results clearly show that adolescents in Lithuanian context start dating very early, at the age of 14 (median age). Secondly, it demonstrates that negative experiences are frequently associated with these early dating experiences. This calls for health promotion action and sexual and mental health literacy programs aimed at young people at early adolescence.

The article is within the journal scope as it discusses adolescents´s sexual behavior and attitudes from perspectives of mental and public health. It also suggests preventive measures as a public health action.

The data could be more comprehensively presented. It was somewhat challenging to understand the structure of the questionnaire and in particular, the response options of the asked items. It would be helpful for the reader, if the items were listed either in the methods or in supplementary material. Also, I would suggest to list the sociodemographic factors clearly in the methods section, i.e. was sexual orientation one of them. Also, response options for the sociodem. items could be provided. Was gender dichotomous (female vs male) and sexual orientation too (heterosexual vs other)? what about gender and sexual diversity? Place of residence was significant. Should it not be adjusted for in the further tests? I think this is also a very relevant point as a few of the findings varied with respect to place of residence. Adjusted models could help interpretation of the results.

The study was scientifically sound and technically fine. The design was cross-sectional but provides valuable and important insight into adolescents´ attitudes and dating behavior and raises the weak points in this sensitive developmental period in young people’s lives, which need to be addressed by educators and/or school health, in collaboration with mental health professionals.

Due to the specific nature of study and target group, the readership may be quite limited. However, the research question is very topical and under-researched and also, the increasing mental health problems among young people are very much in the forefront of societal discussion. Therefore, the interest might be broader than thought at first glance.

This article definitely is an important contribution to the field and more so, due to specific recommendations for preventive interventions. In the light the currently increasing internalizing problems, especially among girls, these findings should be carefully weighed when planning any interventions at youth.

Generally Engllish level is appropriate and understandable. Some refining could be done to improve fluency and make the work more reader-friendly. I have given some suggestions but was under the impression that the specific recommendations with respect to language are provided internally by the journal.

Overall recommendation

Accept after minor revisions.

INTRODUCTION

The introduction is very interesting and generally very clearly written. The study is well motivated, although the aim is quite overall. Specific objectives should be added, as well as hypotheses to be tested. I have one overall linguistic suggestion to rephrase some of the many repeating “Research showed..” in both introduction and discussion.

Line 46: provide background data-building FOR preventive..

Line 44-47: suggest to rephrase and dividing sentence into two. E.g. “Investigations into dating experiences, expectations, and attitudes among young people are necessary… “ In the present form, the sentence is quite heavy and a bit difficult to follow. Also, do the authors mean developmental trajectories of individuals’ lives or changing trends over time at a population level?

Line 48: policy-makers are not meeting > policy-makers do not meet

Line 53: therefore > but

Line 55-56: Change the sentence to: Since building a romantic relationship is new for adolescents, their dates and romantic relationships differ from those of adults.

line 61: Replace EXPERIENCE to after traumatic

line 66-70: There is a bit of repetition, could the two sentences be somehow fused for fluency?

MATERIALS AND METHODS

Materials and methods are generally fine but some more detailed information could be added to help the reader to understand.

Participants and procedures.

On what basis were the six participating schools invited? Convenience sampling? How were the adolescents selected? Were there any inclusion and exclusion criteria? Please indicate.

I would like to see a short passage about the cities. In the findings, there were quite a lot of differences between Vilnius and Klaipeda, so I think it would be relevant to describe them. This would bring the study more into its context.

The authors state that they obtained permissions from the schools and the communities. Was the study approved by any ethics committee? If so, please state the reference number.

In my opinion the procedure was nicely and detailed explained but perhaps it could be restructured a bit to add clarity. The sentence lines 86-87 was a little confusing. Please remove “…’together with explained’..)

line 86-87: Content of the questionnaire was briefly described, together with explained possibility to refuse….  and participants were informed that they had a possibility to refuse participate. This is difficult to understand, please rephrase.

Questionnaires: It would be helpful if you can describe the main items (41 items; 4 sociodemographic items?) of the questionnaire a bit more, including items and response options. Please clearly indicate the sociodemographic factors gender, sexual orientation, place of residence and living with parents with responses. In the results tables, it appears that gender and sexual orientation were dichotomous (female vs male AND heterosexual vs other) was it so, or were multiple response options (e.g, ‘other’ gender or ‘bisexual’, ‘homosexual’ ‘pansexual’) pooled together in the analysis phase?  Think this would be essential information. Especially, if the options were originally dichotomous. Why was this decided so? Sexual and gender diversity is a highly topical issue, and in my opinion a short paragraph discussing this in the paper would fit very well. Authors mention in the discussion (lines 343-344) the gender- and sexual minority, but this doesn´t directly seem to reflect the data or findings (at least in terms of dichotomous gender).

‘Social networks’ are mentioned for the first time in results 3.2. It is difficult to assess the meaning of this if the options are not provided. Alternatively, could you provide the questions as a supplementary material? Include the first sentence of the statistics section here.

Suggest to replace youngster by adolescents for clarity throughout the manuscript.

Statistical analysis: Please see comment above. It would be helpful to building an overall picture of the data, if you first presented the background variables as descriptive statistics (boys vs girls; place of residence; family status; sexual orientation), maybe even just in the text. Then, you could adjust the other tests for the significant background variables?

What statistical program were you using?  I was wondering whether it would be possible to run logistic regression instead of multiple chi square tests.

Non-negative experience sounds somewhat cumbersome. Could it be ‘positive ’ and explained in methods, under the questionnaire?

I suggest you bold all subheadings (Participants and procedure. Questionnaires. ect ) This would help the reader to follow.

RESULTS

Since not all participants responded to all items, please provide missing values or n/N (%-age) when presenting test results throughout the Results section, table headings etc and e.g. as footnotes in tables. Also add N in table headings.

Section 3.2. of results: social networks and psychotropics appear here for the first time. What were the response options? How were social networks defined? Please add to the questionnaire description section of methods. Wondering if older vs younger instead of ‘greater age’ would be more readable.

Tables: Connected to my comment for the statistics section, now the sociodemographic variables are presented in table 1 together with the dating experience. This is a bit difficult to follow and building an overall picture of data is challenging. Maybe you could put dating experience and non-negative vs negative dating experience in one table (adjustments for the significant sociodemographic variables?).

Results section 3.3. Begins with half (51.3%) but no n. Did the entire sample (N=377) answer? 

line 196: Remove statistically significantly and add p-value is brackets.  “Girls, statistically significantly,….” to for example: Gilrs indicated that they would talk about their negative experiences more frequently than boys (. The differences was statistically significant.’

line 199: non-traditional sexual orientation? Please keep to one term.

3.4. Having a partner. Here as well, it would be helpful for the reader to see what the options for having a partner from (what) environments are? Are ‘environments’ as they are referred to here, the same as social networks?

Line 218: Is ‘Dating expectations among adolescents without dating experience’ meant to be a sub-heading? Please bold or make it into a separate sub-section 3.5.

line 222 and throughout the manuscript, please edit ON social networks > IN social networks

line 230 EDIT: Date’s place > dating place / place of dating.

lines 231-233: Please rephrase into a sentence with percentage in brackets: For instance: Other respondents preferred public places, a walk, a café … (54.2%, …. , respectively).

DISCUSSION

It the median age for the first date the most important finding. The authors continue in the second sentence to explaining how this age-related finding contributes to building preventive programs. Although I agree that this study is important for designing such interventions, this statement could be placed a bit more efficiently. Here it feels a bit detached and out of context. Suggest the authors place is after a few more key findings, relating the current results and the conclusive statement more strongly. Generally, I think a clearly stated hypothesis would help to structure the discussion and make it more reader-friendly.

line 245-247: I think the differences between cities could be elaborated a bit more, including the reasons. Please see previous comment for the methods, related to description of cities.

303-306: Logic is a little confusing and difficult to follow. Could authors rephrase to clarify? What were the main messages derived from the data. I think this passage could be better placed in the conclusion or at least towards the end of discussion.

line 317: In this study, we also investigated the people that late adolescents talk with… I don´t think you investigated these people, but adolescents´ perceptions/experiences. Please rephrase.

line 323: peer self-help > peer suppot

line 335-342: Is it possible that psychotropics are for instance more readily available in Vilnius than Klaipeda?

line 343: Please see previous comment about sexual and gender minority. To my understanding, your data do not represent gender minorities at least. And although a great majority of the students were heterosexuals, would it be more neutral to not talk about sexual minority here, but instead, just the heterosexual and others. Again, I strongly recommend to open this topic a bit in the materials and methods with respect to phrasing of the question in the questionnaires etc.

line 343-344: Please see comment above (methods section, referring to same lines)

Limitations:

lines 382-394: Authors state they did not use the available scales, but instead used their own. What are the main benefits and strengths of using the self-developed questionnaire in authors’ opinion? Further, studies using different instruments are not to be compared directly as they measure different aspects. In this sense, I disagree with the authors (in line 387). I would rather state that their data complement the existing findings based on validated scales and also that these findings provide a valuable insight into adolescents’ own subjective experience. Further, would it be feasible to validate the questionnaires, do authors think this study could be repeated in another context, also cross-culturally?

line 388: ‘Research data demonstrated ..’ somewhat gives the impression that the authors talk about their own study. Could be rephrased e.g. previous research / findings or something like that. 

Author Response

Authors point by point responses are attached as a separate file. Many thanks to the Reviewer for the valuable comments. 

Reviewer 2 Report

Very interesting topic discussing dating culture and thank you for submitting it. Here are my comments. 

Introduction

Author states that the literature is coming from a systematic review, which is a whole methodology to gather data. You can say that previous research has shown or research has shown but unless you did a true systematic review, I would not claim it. 

In the literature review, I would also include a sentence about what age range adolescents.

The title should include a more specific population i.e. high school graduates. Also, if you are doing students in high school... the term graduates is misleading. I would then change it to high school students in the title and not school graduates. 

Methods

I would reword this to state.." Inclusion criteria included students that were over the age of 16, etc. "Participants were informed about the aims and scope of the 82 survey, were more than 16 years old, and filled in the paper questionnaires of their own 83 free will. "

Authors should elaborate on how the 45 question instrument was developed. Was it taken from already existing surveys? If not, then why? 

Results

I believe running a Logistic Regression and reporting an Odds Ratio (OR) would be more beneficial to the study. Using this type of analysis will allow you to compare groups within those socioeconomic demographics Example: were men 3x more likely than women to experience dating violence,  were other  89% less likely to experience dating violence than heterosexual, etc. Also, you could report if the model was significant or not. 

Author Response

Authors are very thankful to the reviewer for the comments and have prepared and attached point by point responses as a separate file.

Round 2

Reviewer 2 Report

The author has updated the manuscript with revisions.